# META-LEARNING OF STRUCTURED TASK DISTRIBUTIONS IN HUMANS AND MACHINES

Sreejan Kumar[1], Ishita Dasgupta[2], Jonathan D. Cohen[1,3], Nathaniel D. Daw[1,3], and Thomas L. Griffiths[2,3]

[1]Princeton Neuroscience Institute
[2]Department of Computer Science, Princeton University
[3]Department of Psychology, Princeton University

## ABSTRACT

In recent years, meta-learning, in which a model is trained on a family of tasks (i.e. a task distribution), has emerged as an approach to training neural networks to perform tasks that were previously assumed to require structured representations, making strides toward closing the gap between humans and machines. However, we argue that evaluating meta-learning remains a challenge, and can miss whether meta-learning actually uses the structure embedded within the tasks. These meta-learners might therefore still be significantly different from humans learners. To demonstrate this difference, we first define a new meta-reinforcement learning task in which a structured task distribution is generated using a compositional grammar. We then introduce a novel approach to constructing a "null task distribution" with the same statistical complexity as this structured task distribution but without the explicit rule-based structure used to generate the structured task. We train a standard meta-learning agent, a recurrent network trained with model-free reinforcement learning, and compare it with human performance across the two task distributions. We find a double dissociation in which humans do better in the structured task distribution whereas agents do better in the null task distribution – despite comparable statistical complexity. This work highlights that multiple strategies can achieve reasonable meta-test performance, and that careful construction of control task distributions is a valuable way to understand which strategies meta-learners acquire, and how they might differ from humans.

## 1 INTRODUCTION

While machine learning has supported tremendous progress in artificial intelligence, a major weakness – especially in comparison to humans – has been its relative inability to learn structured representations, such as compositional grammar rules, causal graphs, discrete symbolic objects, etc. (Lake et al., 2017). One way that humans acquire these structured forms of reasoning is via "learning-to-learn", in which we improve our learning strategies over time to give rise to better reasoning strategies (Thrun & Pratt, 1998; Griffiths et al., 2019; Botvinick et al., 2019). Inspired by this, researchers have renewed investigations into meta-learning. Under this approach, a model is trained on a family of learning tasks based on structured representations such that they achieve better performance across the task distribution. This approach has demonstrated the acquisition of sophisticated abilities including model-based learning (Wang et al., 2016), causal reasoning (Dasgupta et al., 2019), compositional generalization (Lake, 2019), linguistic structure (McCoy et al., 2020), and theory of mind (Rabinowitz et al., 2018), all in relatively simple neural network models. The meta-learning approach, along with interaction with designed environments, has also been suggested as a general way to automatically generate artificial intelligence (Clune, 2019). These approaches have made great strides, and have great promise, toward closing the gap between human and machine learning.

However, in this paper, we argue that significant challenges remain in how we evaluate whether structured forms of reasoning have indeed been acquired. There are often multiple strategies that

can result in good meta-test performance, and there is no guarantee a priori that meta-learners will learn the strategies we intend when generating the training distribution. Previous work on meta-learning structured representations do partially acknowledge this. In this paper, we highlight these challenges more generally. At the end of the day, meta-learning is simply another learning problem. And similar to any vanilla learning algorithm, meta-learners themselves have inductive biases (which we term *meta-inductive bias*). Note that meta-learning is a way to *learn* inductive biases for vanilla learning algorithms Grant et al. (2018). Here, we consider the fact the meta-learners themselves have inductive biases that impact the kinds of strategies (and inductive biases) they prefer to learn.

In this work, the kind of structure we study is that imposed by compositionality, where simple rules can be recursively combined to generate complexity (Fodor et al., 1988). Previous work demonstrates that some aspects of compositionality can be meta-learned (Lake, 2019). Here, we introduce a broader class of compositionally generated task environments using an explicit generative grammar, in an interactive reinforcement learning setting. A key contribution of our work is to also develop control task environments that are not generated using the same simple recursively applied rules, but are comparable in statistical complexity. We provide a rigorous comparison between human and meta-learning agent behavior in tasks performed in distributions of environments of each type. We show through three different analyses that human behavior is consistent with having learned the structure that results from our compositional rules in the structured environments. In contrast, despite training on distributions that contain this structure, standard meta-learning agents instead prefer (i.e. have a meta-inductive bias toward) more global statistical patterns that are a downstream consequence of these low-dimensional rules. Our results show that simply doing well at meta-test on a tasks in a distribution of structured environments does not necessarily indicate meta-learning of that structure. We therefore argue that architectural inductive biases still play a crucial role in the kinds of structure acquired by meta-learners, and simply embedding the requisite structure in a training task distribution may not be adequate.

## 2 EMBEDDING STRUCTURE IN A TASK DISTRIBUTION

In this work, we define a broad family of task distributions in which tasks take place in environments generated from abstract compositional structures, by recursively composing those environments using simple, low-dimensional rules. Previous work on such datasets (Lake & Baroni, 2018; Johnson et al., 2017) focuses primarily on language. Here we instead directly consider the domain of structure learning. This is a fundamental tenet of human cognition and has been linked to how humans learn quickly in novel environments (Tenenbaum et al., 2011; Mark et al., 2020). Structure learning is required in a vast range of domains: from planning (understanding an interrelated sequence of steps for cooking), category learning (the hierarchical organization of biological species), to social inference (understanding a chain of command at the workplace, or social cliques in a high school). A task distribution based on structure learning can therefore be embedded into several domains relevant for machine learning.

Kemp & Tenenbaum (2008) provide a model for how people infer such structure. They present a probabilistic context-free graph grammar that produces a space of possible structures, over which humans do inference. A grammar consists of a start symbol $S$, terminal and non-terminal symbols $\Sigma$ and $V$, as well as a set of production rules $R$. Different structural forms arise from recursively applying these production rules. This framework allows us to specify abstract structures (via the grammar) and to produce various instantiations of this abstract structure (via the noisy generation process), naturally producing different families of task environments, henceforth referred to as task distributions.

We consider three structures: chains, trees, and loops. These exist in the real world across multiple domains. Chains describe objects on a one-dimensional spectrum, like people on the left-right political spectrum. Trees describe objects organized in hierarchies, like evolutionary trees. Loops describe cycles, like the four seasons. Here we embed these structures into a grid-based task.

Exploration on a grid is an extensively studied problem in machine learning, particularly in reinforcement learning. Further, it is also a task that is easy for humans to perform on online crowdsourcing platforms – but not trivially so. This allows us to directly compare human and machine performance on the same task. Fig. 1 displays the symbols of the grammar we use and the production rules that give rise to grids of different structural forms.

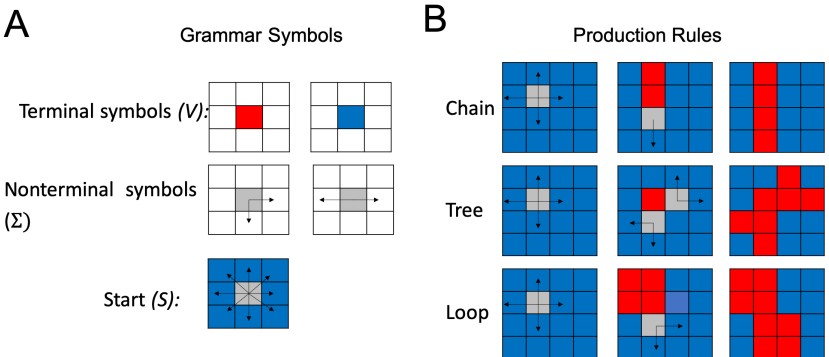

Figure 1: **Generative Grammar** (A) Grammar symbols and (B) production rules. A board is formed by beginning with the start symbol and recursively applying production rules until only terminal symbols (red and blue tiles) are left. Each production rule either adds a non-terminal symbol (from first column to second) or a terminal symbol (from second column to third) with 0.5 probability.

## 2.1 A TASK TO TEST STRUCTURE LEARNING

Here we describe the specific task built atop this embedding of structural forms. We use a tile revealing task on the grid. Humans as well as agents are shown a $7 \times 7$ grid of tiles, which are initially white except for one red tile. The first red tile revealed at the beginning of the episode is the same tile as the initial start tile of the grid's generative process (see Fig. 1). Clicking white tiles reveal them to be either red or blue. The episode finishes when the agent reveals all the red tiles. There is a reward for each red tile revealed, and a penalty for every blue tile revealed. The goal therefore is to reveal all the red tiles while revealing as few blue tiles as possible. The particular configuration of the red tiles defines a single task. The distribution of tasks for meta-learning is defined by the grammar from which these structures are sampled. Here, we randomly sampled from a uniform mixture of chains, trees, and loops as defined in Fig. 1.

## 2.2 A STATISTICALLY EQUIVALENT NULL TASK DISTRIBUTION

Previous approaches to evaluating whether machine-learning systems can extract compositional structure (Lake & Baroni, 2018; Dasgupta et al., 2018) have relied on examining average performance on held-out examples from compositionally structured task distributions. However, we argue that this often confounds whether a system has truly internalized this underlying structure or whether it is relying on statistical patterns that come about as a consequence of compositional rules.

To directly examine whether structured reasoning is a factor in how humans and meta-learning agents perform this task, we need a control task distribution that is similar in statistical complexity, by generating one based on those statistics rather than the direct use of the compositional grammar. To this end, we trained a fully connected neural network (3 layers, 49 units each) to learn the conditional distribution of each tile given all the other tiles on the compositional boards. Note that these conditional distributions contain all the relevant statistical information about the boards. We do this by training on an objective inspired by masked language models like BERT (Devlin et al., 2018). The network was given a compositional board with a random tile masked out and trained to reproduce the entire board including the randomly masked tile. The loss was binary cross entropy between the predicted and actual masked tiles. The network was trained on all possible compositional boards for $10^4$ epochs, and achieved a training accuracy of $\sim$99%.

We then sampled boards from these conditionals with Gibbs sampling. We started with a grid in which each tile is randomly set to red or blue with probability 0.5. We then masked out a tile and ran the grid through the network to get the conditional probability of the tile being red given the other tiles, turning the tile red with that probability. We repeated this by masking each tile in the $7 \times 7$ grid (in a random order) to complete a single Gibbs sweep, and repeated this whole Gibbs sweep 20 times to generate a single sample. We refer to the distribution of boards generated this way as the null task distribution. Fig. 2 shows example compositional and null distribution grids.

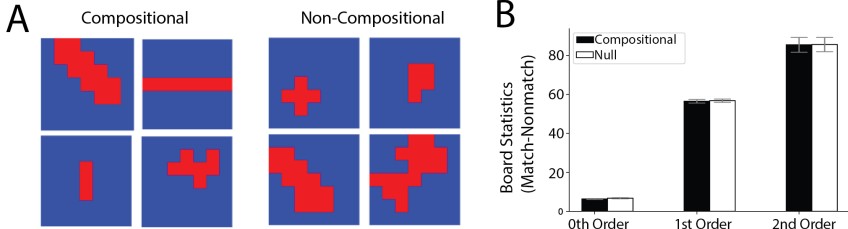

Figure 2: **Comparing compositional and null task distributions** (A) Example compositional and null distribution boards. Compositional boards are distinctly either chain, trees, or loops while null boards have similar statistical properties but don't necessarily obey the recursive rules used to generate compositional boards. (B) Ising statistics across the two task distributions. Error bars are 95% non-parametric bootstrap confidence intervals across different boards of the respective distribution.

While the statistical structure looks similar, the non-compositional null boards shown could not have been generated by the grammar in Fig. 1. The conditional distributions for the two distributions are similar by design; we further quantify statistical similarity using Ising statistics (Zhang, 2007). We compared the 0th order, 1st order, and 2nd order effects defined as follows. The 0th order statistic corresponds to the number of red minus number of blue tiles. The 1st order statistic counts the number of agreeing neighbours (vertically or horizontally adjacent) minus the disagreeing ones, where agreeing means being of the same color. The 2nd order statistic is the number of triples (tile + its neighbor + its neighbor's neighbor) that agree, minus those that don't. Fig. 2b shows that the two distributions are not significantly different in terms of the Ising statistics measured ($p > 0.05$ for all three orders).

The principal difference between these two task distributions is the way in which they were generated. The compositional task distribution was generated through the recursive application of simple, low-dimensional rules that generates a mixture of three discrete structures, whereas the null task distribution was generated through a more complex Gibbs sampling procedure that is not explicitly compositional and does not utilize explicit simple, low-dimensional rules. Although it is true that some boards within the null task distribution may be consistent with a simple compositional grammar, the distribution as a whole was not generated through a compositional grammar.

## 3    EXPERIMENTS

We analyze and compare the performance of standard meta-learners and human learning on our tile-revealing task. We test them on boards that are sampled from the generative grammar and contain explicit compositional structure, as well as on boards that are matched for statistical complexity, but are sampled from a null distribution that was constructed without using explicit compositional structure. Comparing performance across these two task distributions allows us to pinpoint the role of simple forms of structure as distinct from statistical patterns that arise as a downstream consequence of compositional rules based on such structure.

### 3.1    METHODS

**Meta-Reinforcement Learning Agent**    Following previous work in meta-reinforcement learning (Wang et al., 2016; Duan et al., 2016) we use an LSTM meta-learner that takes the full board as input, passes it through 2 fully connected layers (49 units each) and feeds that, along with the previous action and reward, to 120 LSTM units. It is trained with a linear learning rate schedule and 0.9 discount. The reward function was: +1 for revealing red tiles, -1 for blue tiles, +10 for the last red tile, and -2 for choosing an already revealed tile. The agent was trained using Advantage Actor Critic (A2C) (Stable baselines package Hill et al., 2018). The agent was trained for $10^6$ episodes. We performed a hyperparamater sweep (value function loss coefficient, entropy loss coefficient, learning rate) using a held-out validation set for evaluation (see Appendix). The selected model's performance was evaluated on held-out test grids. We trained different agents in the same way on the compositional and null task distributions, with separate hyperparameter sweeps for each.

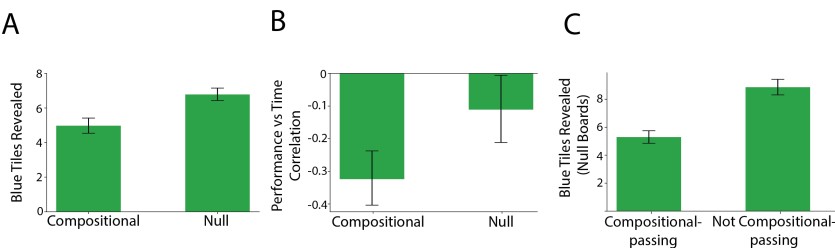

Figure 3: **Human performance.** (A) Humans perform better (i.e. less blue tiles) in the compositional vs null task distribution (p<0.0001). (B) Human performance improves over the course of the experiment (indicated by negative correlation between trial number and number of blue tiles revealed), with significantly greater improvement for compositional distribution (p=0.0006). (C) Some null distribution boards can pass as compositional – humans perform significantly better on these than on other boards in the null distribution (p<0.0001). Agents (not shown in plot) do not do significantly differently on these boards (p>0.05).

**Human Experiment** We crowdsourced human performance on our task using Prolific (www.prolific.co) for a compensation of $1.50. Participants were shown the $7 \times 7$ grid on their web browser and used mouse-clicks to reveal tiles. Each participant was randomly assigned to the compositional or null task distribution, 25 participants in each. Each participant was evaluated on the same test set of grids used to evaluate the models (24 grids from their assigned task distribution in randomized order). Note that a key difference between the human participants and model agents was that the humans did not receive training on either task distribution. While we are interested in examining whether agents can *meta-learn* abstract structures (by training on compositional task distributions), we assume that humans already have this ability from pre-experimental experience. Since participants had to reveal all red tiles to move on to the next grid, they were implicitly incentivized to be efficient (clicking as few blue tiles as possible) in order to finish the task quickly. We found that this was adequate to get good performance. A reward structure similar to that given to agents was displayed as the number of points accrued, but did not translate to monetary reward.

**Evaluation** Unless specified otherwise, performance is evaluated as the number of blue tiles revealed before all red tiles are revealed (lower is better). All error bars are 95% non-parametric bootstrap confidence intervals calculated across agents / participants. Non-overlapping confidence intervals will have a significant difference, but we also include non-parametric bootstrapped p-values for differences across different samples (e.g. human vs agent).

## 3.2 RESULTS

In this section, we first describe human behavior on this novel task. We see that humans perform better on the compositional distribution, without extensive training and even while directly controlling for statistical complexity. We then compare human performance with that of a meta-learning agent—which has had extensive training on this task, and therefore has had the chance to learn the structure relevant to this task distribution. We find significant qualitative and quantitative differences in behavior, and examine the role of meta-inductive bias – i.e. what kinds of cross-task structure do meta-learners prefer to represent? In particular, we consider compositional and spatial structure. Finally, we demonstrate the effect of an architectural change (adding convolutions) in the meta-learner that makes it easier for it to discover spatial structure. We demonstrate that, while this helps agent performance overall, it further highlights the divergence between human and agent behavior in learning the compositional rule-based structure in our task distributions.

**Human performance:** We found that participants perform better on the compositional task distribution than the null task distribution (see Fig. 3a). Despite not having been trained on this task beforehand, human participants do fairly well on this task from the outset, suggesting that humans might have some of the relevant inductive biases from pre-experimental experience. To test if there is learning within the experiment itself, we correlated trial number with the number of blue tiles revealed (Fig. 3B), and found improvement across both conditions but significantly greater im-

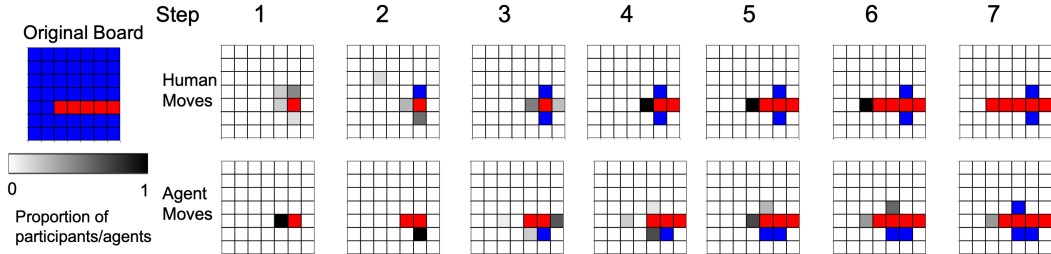

Figure 4: **Human and agent policies on the task.** Red/blue indicate already revealed tiles while grayscale indicate what proportion of humans or agents (conditioned on the moves so far) revealed that tile in the next step. In this chain example, once humans figure out that the board is a chain structural form (step 5), they exhibit perfect performance by choosing tiles along the chain's production direction, while agents still choose other blue tiles.

provement for the compositional distribution. Finally, we investigate performance on the null task distribution more closely. There is some overlap between the null and compositional distributions, because some of the generated null boards could have been generated by the same exact production rules of the generative grammar for the compositional task distribution. We split the null test set by whether or not the board is 'compositional-passing' and compare human performance across these. To do this, we generated the set of all possible compositional boards on a $7 \times 7$ grid and labeled any null task distribution board as compositional-passing if it happened to be a part of this set. We find that humans do significantly better on boards that could have have been generated by the compositional production rules (Fig. 3c). This further suggests recognition and use of low-dimensional rules that align more closely with the compositional distribution than the null distribution.

**Comparing human and agent performance:** First, we note that the meta-learners perform relatively well on this task (Fig. 5), indicating that they have learned some generalizable information from the distribution of tasks. Since the test set has held-out boards of a compositional grammar, this might be taken as evidence that the agents discovered the compositional structure used to generate the boards. Here, we attempt to decouple this possibily – that is, that agents learn to infer and use the simple, low-dimensional compositional rules as humans appear to do – from the possibity that agents learn statistical patterns that are a *consequence of* the use of compositional rule.

We start with an example, involving the chain structure, that highlights the difference between human and agent policies on this task (Fig. 4). In this example, once humans figure out that the board is a chain structural form, they never deviate from the chain's production direction while agents do. This suggests that humans are learning the simple generative rule of the chain form and using this rule to determine their actions, while the agent is using statistical patterns associated with the chain rule rather than the rule itself.

We now consider various ways to quantify this difference. First, we see that humans do better overall on both the compositional and null distributions (Fig. 5; p<0.0001 for both task distributions). This is despite, unlike the agents, having no direct experience with this task. This suggests that humans have useful inductive biases from pre-experimental experience that are valuable in this task (Dubey et al., 2018); for example, the tendency to recognize and utilize low-dimensional, composable rules, and the tendency to look for spatial patterns. We discuss the role of these inductive biases in the following sections. The meta-learner has had extensive experience with each task distribution, and had the chance to discover the structure / inductive biases relevant for this task. The differences in performance indicate that standard meta-learners differ from humans in the kinds of structure / inductive biases they learn (i.e. in their meta-inductive biases).

**Bias toward simple discrete rules** First, we note that humans perform better on the compositional versus the null distribution (Fig. 5a), whereas the agent does better on the null task distribution than on the compositional tasks. This reflects a notable difference between their performance. We hypothesized that humans perform well on the compositional task distribution by first inferring what kind of structure is relevant for the current task, and then following the production rules for that

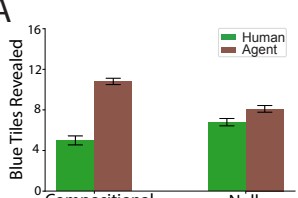
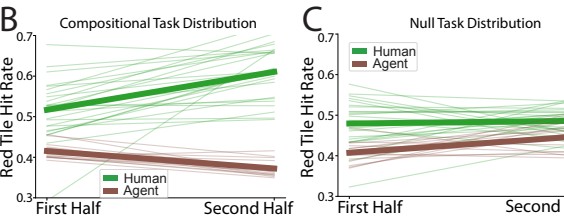

Figure 5: **Comparing human and agent performance** (A) Humans do better at the compositional task than the null (p<0.0001), while agents do better at null (p<0.0001). (B) Humans have a higher success rate revealing red tiles in the second half of a trial for the compositional task (p<0.0001), agents do not. Transparent line represents individual human/agent average over trials, thick lines represent average over humans/agents. (C) Humans do not improve their success rates during a trial in the null task while agents do (p=0.0014).

structure. Since such structure was not used to create the null distribution, the act of learning, inferring, and using this structure is not as helpful in the null task distribution. Further, we hypothesized that the agents learn statistical patterns instead.

Fig. 4 supports this intuition but here we look to quantify it further. If a system represents a set of discrete classes of structures corresponding to our compositional rules, we would expect success rate (rate of choosing red tiles) to be low in the beginning of a trial while figuring out what the structure underlying this trial is. Conversely, we would expect a higher success rate towards the end while following inferred production rules to reveal red tiles. To test this hypothesis, we split human and agent behavior in each trial into the first and last half, and examine success rate in each Fig. (5b and c). For the compositional distribution, we find that humans have a higher success rate in the second half, providing support for our hypothesis. In contrast, we find that agent success rate does not increase over a trial, and in fact decreases. We also find that humans do not show increasing success rate in the null task distribution while agents do, providing further evidence for our hypothesis.

**Bias toward spatial proximity.** We note that humans outperform the agent even in the null task distribution, despite extensive training for the agent. One possibility is that good human performance in the null task is explained by their performance on the compositional-passing examples in the null task distribution (Fig. 3c). However, another possibility is that humans come to the task with strong inductive biases about spatial proximity.[1] While the starting tile for the grammar can be randomly chosen, the production rules operate over nearest-neighbour adjacencies. A system that has a bias toward local spatial structure might therefore perform better at the task.

We test this possibility by comparing performance to a heuristic that uses only local spatial information. This heuristic selects uniformly from the (unrevealed) nearest neighbors of a randomly selected red tile. We evaluated this heuristic 1,000 times on each test board and formed a z-score statistic by subtracting the mean heuristic performance from the human's/agent's performance for each board divided by the standard deviation of the heuristic's performance. We find that humans do better than the neighbor heuristic (Fig.6a), while the agent does not. This indicates that humans' inductive bias for spatial proximity may partially explain the differences in performance across humans and agents.

We can give a neural network a bias toward spatial proximity using convolutions (LeCun et al., 1989). To test if this helps the agent, we replaced the agent's first fully connected layer with a convolutional layer. We find that this agent outperforms humans on the null task distribution(Fig.6b). We also find that it outperforms the spatial heuristic described above (Fig.6c). Note that this strictly reduces the expressivity (i.e. the number of parameters) of the model, and any improvements are due to the right meta-inductive bias (i.e. the right architectural inductive bias given to the meta-learner). However, humans still perform better than the agent in the compositional task distribution. Crucially, this means that which distribution is easier is different for the human and the agent – humans find the compositional tasks easier than null, while the agent finds the null tasks easier than the

---

[1]Spatial structure is shared by both distributions (Fig. 2) and can't explain why humans are better at compositional tasks while agents are better at null. However, here we investigate whether it can explain why humans perform better *overall*.

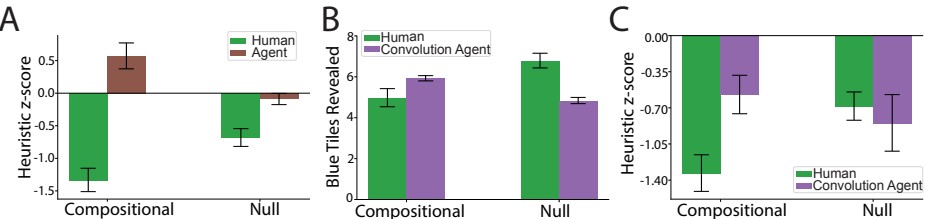

Figure 6: **Role of spatial structure in task performance.** (A) Humans outperform the neighbor heuristic (negative z-score), the agent performs worse. (B) Humans outperform the convolution agent in the compositional distribution (p<0.0001) and the convolution agent outperforms humans in the null distribution (p<0.0001). (C) The convolution agent outperforms neighbor heuristic.

compositional. This result exhibits a double dissociation between learning simple, abstract structure and statistical learning. It shows that the gap between humans and agents on the compositional task is not due to artificial meta-learners being overall worse learners than humans – the convolutional meta-learner actually outperforms humans on the null distribution of equal statistical complexity. This provides further evidence that the inductive bias toward and representing of simple abstract structure is what may give humans a competitive advantage over the agent on the compositional task distributions, and that meta-learners do not learn it despite access to compositional training distributions.

## 4 DISCUSSION

The ability to recognize structure in environments, as well as learn and utilize this structure, is a central tenet of human intelligence (Lake et al., 2017). One example of this kind of structure is compositional grammars. These use of simple, low-dimensional rules that can be recursively applied to produce arbitrary complexity and can be widely generalized outside the training distribution. An inductive bias toward structured representations could be of great value to machine learning systems. Recent developments in meta-learning hold promise as an approach to endowing systems with such useful inductive biases. In this work, we make several methodological and scientific contributions to provide a rigorous way to test for structured forms of reasoning using compositional grammars as a case study. We show that human behavior is consistent with learning and utilizing low-dimensional compositional rules. We also show that standard meta-learning approaches, in sharp contrast to humans, struggle with discrete abstract structures and prefer statistical patterns.

Our first contribution is the development of compositionally structured, rule-based task distributions for meta-learning using explicit generative grammars (Kemp & Tenenbaum, 2008). Previous work on generating compositional datasets has focused on language. We argue that using explicit generative grammars has the dual advantage of being generalizable to a variety of structures, as well as being easy to embed in multiple domains relevant to machine learning. In this work, we embed this structure into a grid-based task. Grid-based tasks are commonly studied in reinforcement-learning, are easy for humans to perform on online platforms, and behavior on this task is easy to visualize, analyze, and interpret. This provides fertile ground for direct comparisons between human and machine behavior, as we demonstrate in our experiments. Previous work on meta-learning compositionality uses performance on a compositional task distribution as an indicator for meta-learning this structure (Lake, 2019). However, we show that it is possible for meta-learning systems to perform well using statistical patterns instead.

Our second methodological contribution is to create distributions with comparable statistical complexity to the structured distribution, that do not directly use the rules used to form the structured distribution. This control distribution allows us to disentangle statistical pattern matching from structured reasoning (which, in this specific case, is rule-based compositionality) and highlights the difference between actually learning and utilizing simple abstract structures (e.g. low-dimensional compositional rules) versus using the statistical patterns that may be a downstream consequence of those structures. Our method closely approximates the global statistics that emerge from the compositional rules, by using a neural network to learn the conditional distributions and generating Gibbs

samples from these conditionals. This approach is similar to masked language modelling (Devlin et al., 2018), and our findings—that this procedure generates statistically similar but not explicitly compositional distributions, that are in fact *easier* for downstream networks to learn than the true compositional distribution—are also relevant to understanding the representations learned by these systems more broadly (Rogers et al., 2020).

In our experiments, we first show that humans have a bias toward using the compositional rule-based structure, while directly controlling for statistical complexity. This generalizes findings in the space of function learning (Schulz et al., 2017) to grid-based reinforcement learning tasks. Further, we find that agents (recurrent network trained with model-free reinforcement learning, following Wang et al., 2016; Duan et al., 2016) find the non-compositional null distribution easier to learn than the compositional one. This is in direct contrast with human behavior, indicating that agents do not learn the same strategies that humans use through meta-learning. A followup experiment with a convolutional agent directly dissociates the effectiveness of statistical learning from the inductive bias toward compositional rules, and highlights learning and use of these simple, low-dimensional rules as the key difference between humans and agents in this task. In both sets of experiments, we find a double dissociation between humans and agents: humans find the compositional task easier than the null task, while the pattern swaps for the agent. This indicates a significant difference (orthogonal to overall performance) between the kinds of strategies humans and agents use to solve this task. Our results therefore indicate that learning abstract structure, such as explicit compositional rules, remains *difficult* for artificial meta-learners – and that they prefer other statistical features when possible. In other words, they do not have a meta-inductive bias toward learning low-dimensional compositional rules.

Although the architecture we investigate here does not successfully meta-learn the ability to recognize and use abstract compositional rules, our point is not that this inductive bias cannot be meta-learned. Rather, it is that every meta-learning architecture has its own meta-inductive bias, and we show a specific case in which a standard and widely-used architecture's meta-inductive bias leads to encoding statistical features rather than the low-dimensional compositional rules used to generate the task distribution. When setting out to meta-learn a structured representation using a meta-learning system, it is important to consider the meta-inductive bias of that system in addition to engineering its task distribution. Graph neural networks (Battaglia et al., 2018), neurosymbolic approaches (Ellis et al., 2020), as well as attention mechanisms (Mnih et al., 2014), permit abstraction by (implicitly or explicitly) decomposing the input into parts. Using these in meta-learning architectures might favor structured representations and reasoning.

Although we encourage exploring the space of architectural changes to give meta-learning agents a better chance to learn such structured forms or reasoning, it may also be true that simpler architectures can acquire relevant inductive biases if given a rich enough high-dimensional training environment that rivals the environment(s) in which the species has evolved and individuals learn (Hill et al., 2019). However, the amount of data required to acquire these structured forms of reasoning in "vanilla" architectures may be prohibitively large and largely infeasible. Further, even as training within these extraordinarily large environments becomes more feasible, the biases of the correspondingly large networks being used may continue to affect the ease at which they can learn structured representations. Therefore, it is still worthwhile to investigate the role of architectural modifications on the ability to meta-learn structured representations in smaller environments, such as the ones we present here, so that one day we may transfer those insights to training on larger, more naturalistic environments. An exciting direction for future work is to examine a range of approaches to learning structured representations with the tools we set forth in this paper, and using the resulting insights to move toward closing the gap between human and machine intelligence.

## 5  ACKNOWLEDGEMENTS

We thank Erin Grant for providing helpful comments on the initial version of the manuscript. S.K. is supported by NIH T32MH065214. This work was supported by the DARPA L2M Program and the John Templeton Foundation. The opinions expressed in this publication are those of the authors and do not necessarily reflect the views of the John Templeton Foundation.

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

## A  APPENDIX

### A.1  HYPERPARAMETER DETAILS FOR REINFORCEMENT LEARNING

We did a hyperparameter search for the following: value function coefficient, entropy coefficient, and learning rate. In particular, we evaluated each set of hyperparameters on a separate validation set, selected the model with the highest performing set, and re-trained the model to be evaluated on a previously-unseen test set of boards. Note that the final test set is not seen by the model until the last, final evaluation step. Searches were ran independently for both task distributions (compositional and null). The final selected hyperparameters for both task distribution were: value function coefficient=0.000675,entropy coefficient=0.000675, learning rate=0.00235.

### A.2  DESCRIPTION OF COMPOSITIONAL GRAMMAR

Here we provide an intuitive description of all the compositional grammar rules showin in Fig. 1.

**Start Tile**  Each grammar begins with a start square somewhere on or after the 3rd column/3rd row of the 7x7 grid (so a grammar cannot start with a tile on the first or second row/column of the grid).

**Probabilistic Production** Each grammar rule corresponds to a particular abstract structure. These structures can vary in size based on how many times that grammar rule is applied. Whenever a grammar rule is applied, the grammar will terminate with probability $p = 0.5$. If a grammar rule cannot be applied again (e.g. the current tiles are on the edge and the next production would go off the 7x7 board), then the grammar automatically terminates.

**Chain Production** On the first chain production, the next two red tiles after the start tile $(s_x, s_y)$ will be either: $(s_x-1, s_y), (s_x+1, s_y)$ or $(s_x, s_y+1), (s_x, s_y+1)$. Any subsequent chain productions after the first one will follow the direction of the first production (for example: if the first chain production places red tiles on $(s_x+1, s_y), (s_x+1, s_y)$, the second would do $(s_x-2, s_y), (s_x+2, s_y)$.

**Tree Production** On the first tree production, the next two red tiles will either be $t_1 = (s_x + 1, s_y), t_2 = (s_x, s_y - 1)$ or $t_1 = (s_x + 1, s_y), t_2(s_x, s_y - 1)$ or $t_1 = (s_x - 1, s_y), t_2 = (s_x, s_y + 1)$ or $t_1 = (s_x - 1, s_y), t_2 = (s_x, s_y + 1)$. The tree production rule always builds in two orthogonal directions. On subsequent tree productions, one of the two added red tiles from the previous production will be picked and two orthogonal directions will be picked for the next two red tiles. The defining characteristic in the tree structure is the "lack of loops", which means there can never be a 2x2 sub-square of all red tiles. Therefore, a currently red tile $t$ is chosen for the center of production such that there does exist a pair of tiles $t_1, t_2$ in orthogonal directions to $t$ such that making both $t_1, t_2$ red does not create a 2x2 sub-square of red tiles.

**Loop Production** On the first loop production, a 2x2 red sub-square will form in one of four directions by coloring three tiles surrounding the start square $(t_1 = (s_x + 1, s_y), t_2 = (s_x + 1, s_y + 1), t_3 = (s_x, s_y + 1)$ or $t_1 = (s_x - 1, s_y), t_2 = (s_x - 1, s_y - 1), t_3 = (s_x, s_y - 1)$ or $t_1 = (s_x - 1, s_y), t_2 = (s_x - 1, s_y + 1), t_3 = (s_x, s_y + 1)$ or $t_1 = (s_x + 1, s_y), t_2 = (s_x + 1, s_y - 1), t_3 = (s_x, s_y - 1)$. The next production rule will form another 2x2 red square surrounding an adjacent tile to the original 2x2 red square such that the new 2x2 red square only shares one edge with the old 2x2 red square (see Fig. 1 and 2 for what this exactly looks like).

### A.3 REWARD OF AGENTS OVER TRAINING EPISODES

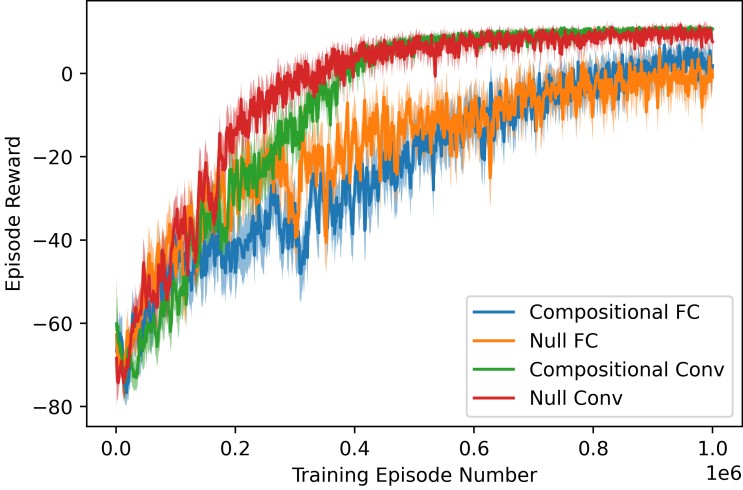

Figure 7: Reward received by each of the four agents trained in this work over the course of the million training episodes.

### A.4 ALL PERFORMANCE DIFFERENCES ACROSS HUMANS AND AGENTS FOR ALL CONDITIONS

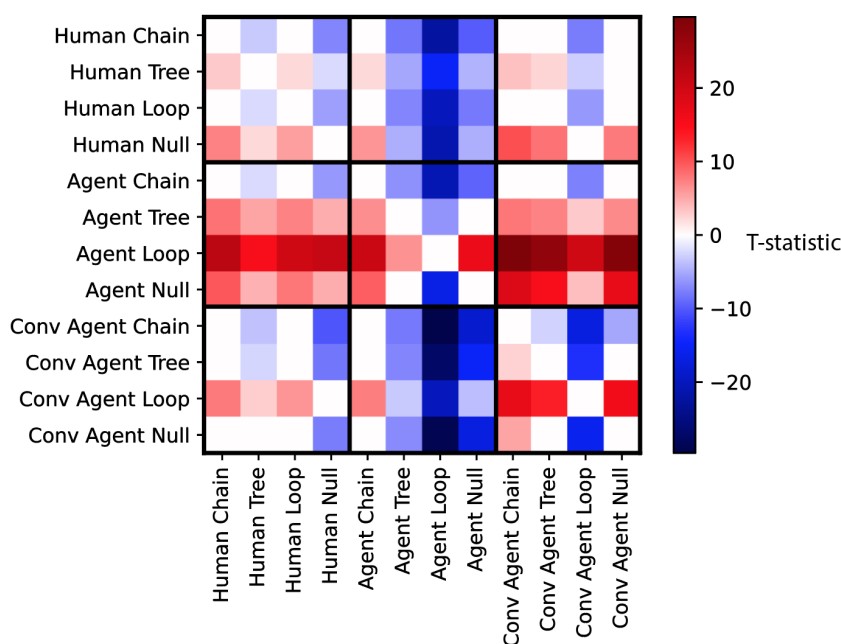

Figure 8: We show the differences in performance (mean number of blue tiles, so lower is better) across humans and agents for the chain, tree, loop compositional conditions and the null condition. Differences are shown as t-values from testing difference in means across different participants or different agent runs. Any non-statistically significant differences are set to 0 (shown as the color white). Note that a negative t-value indicates better performance, since the metric is mean number of blue tiles revealed.

