# OpenReview forum: "Meta-Learning of Structured Task Distributions in Humans and Machines"
_ICLR.cc/2021/Conference — ICLR 2021 Poster_

### Official Review · AnonReviewer3 · 2020-10-15
**Interesting tasks, but model limitations affect the conclusions**

**Rating:** 7
**Confidence:** 5

**Review:**

Post revision update
-------------------------------

The authors have been very helpful and addressed many of my concerns, and I think the revised paper is a substantial improvement. I have rated the paper as a 7, although I do have some lingering concerns.

Most crucially, it's still not clear to me that the compositional rules the authors highlight are the correct way to characterize the differences in patterns of behavior, since, for example, both models significantly outperform humans at the tree rule. However, the authors do point out that humans perform better at these tasks than the null distribution. Still, I worry that the authors are focusing on the wrong dimension along which the compositional and null task distributions differ. However, I think the fact that the authors followed the suggestion to include the results in the appendix is helpful in this regard, at least future researchers will be able to see the full pattern of results to draw their own conclusions.

Original Review
---------------------------

This paper proposes to explore whether meta-learning approaches can exploit a compositional structure in their tasks to generalize, and compares this ability to humans. To do so, the paper introduces a grid dataset consisting of generative grammars for generating compositional grids, as well as a null task distribution which is non-compositional but matches on certain low-order statistics. These tasks are interesting and new. They have humans and agents perform a task to reveal rewarding squares on the grid, and compare to agents that meta-learn this task. Human subjects perform better at the compositional distribution, whereas models perform better at the null distribution. They conclude that "compositional structure remains difficult for these systems and that they prefer other statistical features" and that this "highlights the importance of endowing artificial systems with this bias." While the topic is timely, and the tasks are interesting, there are a number of limitations to the model and training which I think seriously limit the conclusions. I think that the task is really only compositional for a model which is able to fixate on different locations on the grid. Thus, I recommend rejecting for now, although I think a revision with a more sophisticated model and more thorough discussion could be a valuable contribution.


Strengths:

* Interactive tasks for humans and models are a great improvement over prior toy datasets on compositionality, e.g. SCAN.

* I think the tasks are very interesting, and offer directions that aren't really address by prior compositional generalization datasets that mostly rely on composition of words, or on composition of visual properties like color and shape.

* It's great to see actual comparisons to human performance, and careful thinking about how to evaluate compositional generalization vs. other types (though this could be developed further, see below).


Areas for improvement:

* Is generalization equally good for humans under each rule type? If not, this might affect the conclusions. For example, perhaps humans would be very good at inferring chains, but not the more complex structures (like trees). If so, it could be that "compositionality" per se is not the construct underlying their performance, but rather "chains" or some other, simpler construct. Because performance is not presented broken down by structure type, it is difficult to determine whether some pattern like this could explain the results. Thus, it is difficult to conclude that compositionality is the factor underlying the results.

* There are a number of features of the agent that confound the comparison with the humans. These limit the ability to draw a strong conclusion about e.g. "the importance of endowing artificial systems with [a compositional] bias."
    * Why are the main comparisons run using the non-convolutional model, when the convolutional one is clearly more closely matched to humans (as the discussion acknowledges)? It seems like most (although not all) the difference in performance is due to spatial bias, rather than compositionality.
    * Indeed, this spatial bias is partially addressed on the input (by the convolutional experiments), but it is *never* addressed on the output. Humans know that there is a spatial structure to the tiles they are clicking on, the agent can access this information only implicitly.
    * To address this, one could build a recurrent attention model (e.g. Mnih et al, 2014; Gregor et al, 2015) which can make visual saccades around the grid before deciding whether to reveal the square at the current point of fixation, or whether to fixate to another location. This would likely match the human process better, since the humans are likely fixating their gaze on the locations they are considering, rather than fixating in the center of the grid without moving their eyes. It's also motivated by the observation that agents generalize better if they receive ego-centric input rather than visual input fixed on the grid (c.f. Hill et al, 2020; Ye et al, 2020). This is an important issue to the claims at stake. For an agent which could fixate on each location it was considering, the compositional rules would be much more consistent than for an agent that perceives the whole grid from a fixed perspective. For a fixating agent, the compositional rules would also be much more consistent than the null distribution. In fact, I would suggest that it is *only from the perspective of a model which can fixate that this distribution can be considered compositional at all.* How can we tell that the difference between the humans and the model isn't due to the human ability to fixate, rather than some abstract bias toward compositionality?
    * The paper may not be able to address all the ways in which the model's experience of the task is unlike humans, but then the there should be a *corresponding tempering of the conclusion that the comparison to humans says something specific about the difference between the model and humans.* That is, given the current experiments, the discussion of this paper should focus at least as much on the limitations of the present model as on general conclusions about failures of the model class and the need for additional inductive biases.

* Furthermore, exploring compositionality in toy tasks can be misleading. Hill et al. (2020) show that compositional generalization is significantly improved in more realistic settings (for example an RL agent that executes actions over time achieves 100% compositional generalization on a task that a feed-forward classifier only achieves 80% generalization on). They argue that toy stimuli remove one of the most important elements for training deep models — the rich environments in which humans, also, are trained. Even if the input and output of the model and humans were better matched on this dataset, it may be misleading to conclude something as general as "the importance of endowing artificial systems with this [compositional] bias" without giving these models training on a distribution of stimuli and tasks that more closely match the rich variety which humans experience over development. Of course, it is not feasible in practice to do so (yet). But this limitation and its relevance to the conclusions should at least be acknowledged in the discussion.

* The paper could use some more discussion of the distinction between statistical patterns and compositional rules. This seems like an important point, but it wasn't entirely clear to me. Compositional rules correspond to a certain statistical distribution over grids. The fact that up to 2nd order Ising statistics are matched does not mean that the distributions of outcomes are matched "statistically," it merely means they match in certain low-order statistics. It's not clear if these are the right statistics by which to compare the distributions (especially for the non-convolutional model, which has no spatial awareness). The paper would be strengthened by justifying the choice of these statistics more carefully, and articulating a clear distinction between what counts as a "statistical" pattern vs. a rule.





References
---------

Gregor, Karol, et al. "Draw: A recurrent neural network for image generation." arXiv preprint arXiv:1502.04623 (2015).

Hill, Felix, et al. "Environmental drivers of systematicity and generalization in a situated agent." International Conference on Learning Representations, 2020.

Mnih, Volodymyr, Nicolas Heess, and Alex Graves. "Recurrent models of visual attention." Advances in neural information processing systems. 2014.

Ye, Chang, et al. "Rotation, Translation, and Cropping for Zero-Shot Generalization." arXiv preprint arXiv:2001.09908 (2020).

---

> ### Author Response · Authors · 2020-11-22
> **We agree with this review and have significantly reframed the manuscript and briefly tested out the suggested models [1/5]**
>
> We thank the reviewer for their helpful comments. Below is our point-by-point by response to their comments along with relevant manuscript changes.
>
> > Is generalization equally good for humans under each rule type? If not, this might affect the conclusions. For example, perhaps humans would be very good at inferring chains, but not the more complex structures (like trees). If so, it could be that "compositionality" per se is not the construct underlying their performance, but rather "chains" or some other, simpler construct. Because performance is not presented broken down by structure type, it is difficult to determine whether some pattern like this could explain the results. Thus, it is difficult to conclude that compositionality is the factor underlying the results.
>
> We agree that our framing of human performance being well explained by “compositionality” has not been adequately hashed out. We believe humans are significantly better than standard machine-learning methods at task distributions that require recognizing abstract structure (e.g. causal graphs, compositional grammars, discrete symbolic objects, etc). We argue that even the “simplicity” of a chain is due to the simplicity of the underlying rule that generates it (to recursively extend in opposite directions).
>
> As suggested by the reviewer, we do find that humans do better on chains than trees (p<.05 t-test) and on loops than trees (p<.05 t-test). The overarching message we convey is that this very notion of simplicity is based on a structured representation -- i.e. representation in the form of simple recursively applied rules. This is as opposed to the null distribution, which has similar global statistics but is not “simple” in a human sense since it is not underlied by simple recursively applied rules. We agree that the term “compositionality” is confusing in this context and have significantly reframed the manuscript to instead focus more broadly on meta learning approaches to structured representations, with compositionality as a special case.
>
> > Why are the main comparisons run using the non-convolutional model, when the convolutional one is clearly more closely matched to humans (as the discussion acknowledges)? It seems like most (although not all) the difference in performance is due to spatial bias, rather than compositionality.
>
> We agree with the reviewer that the convolutional model is more closely matched to humans. The order of models/comparisons done in this work was mostly to incrementally show the value of adding architectural changes to support inductive bias. A key point in our paper is that “good” performance on meta-test does not indicate that the structure embedded in the task distribution have been internalized by the model or that a standard meta-learner is using these. There are many ways to get decent meta-test performance (in this case, the use of global statistics), not just the structure intentionally embedded during training e.g. spatial or compositional structure.
>
> Just as adding convolution makes it easier for the agent, we want to highlight the importance of considering architectural changes that give the meta-learner a compositional inductive bias as well. This ordering also allows us to highlight that simply performing “better or worse” than humans is not the key take-home message from our paper, but that humans and agents perform differently. Of two statistically similar distributions, humans find the one that was generated with low-dimensional composable rules easier, while agents without explicit compositional bias find the other one easier.
>
> > Indeed, this spatial bias is partially addressed on the input (by the convolutional experiments), but it is never addressed on the output. Humans know that there is a spatial structure to the tiles they are clicking on, the agent can access this information only implicitly.
>
> We first would like to note that, even though spatial proximity is not built-in into the action space, the agent does indeed choose actions close to revealed red tiles. An analysis of the actions of the agent shows that the mean Manhattan distance of an agent’s action and the most recently revealed red tile has a 95% bootstrap confidence interval (confidence interval over different agent runs) is 1.53-1.75. The fact that this distance is significantly below 2 is a statistic showing the agent’s tendency to click on tiles close to the currently revealed red tile, showing that the agent learns to take spatially-proximal actions despite this not being directly told which actions are spatially proximal.

---

> > ### Author Response · Authors · 2020-11-22
> > **We agree with this review and have significantly reframed the manuscript and briefly tested out the suggested models [2/5]**
> >
> > > To address this, one could build a recurrent attention model (e.g. Mnih et al, 2014; Gregor et al, 2015) which can make visual saccades around the grid before deciding whether to reveal the square at the current point of fixation, or whether to fixate to another location. This would likely match the human process better, since the humans are likely fixating their gaze on the locations they are considering, rather than fixating in the center of the grid without moving their eyes. It's also motivated by the observation that agents generalize better if they receive ego-centric input rather than visual input fixed on the grid (c.f. Hill et al, 2020; Ye et al, 2020). This is an important issue to the claims at stake. For an agent which could fixate on each location it was considering, the compositional rules would be much more consistent than for an agent that perceives the whole grid from a fixed perspective. For a fixating agent, the compositional rules would also be much more consistent than the null distribution. In fact, I would suggest that it is only from the perspective of a model which can fixate that this distribution can be considered compositional at all. How can we tell that the difference between the humans and the model isn't due to the human ability to fixate, rather than some abstract bias toward compositionality?
> >
> > We greatly appreciate the reviewer’s insight here. In fact, we strongly agree that fixation would help us tremendously-- in fact we think that allowing for fixation might precisely be the compositional inductive bias required here. Fixation permits us to see the generation process as the recursive application of the same (composable) rule over and over, just with a different fixation point. Being able to fixate properly in this task would therefore be akin to learning the underlying generative grammar rules and following their generative direction.
> > We did investigate utilizing the “Selective Attention Model” in Gregor et al. 2015 (the paper mentioned by the reviewer). In particular, we gave the agent the ability to fixate on a particular subset of the board by applying a gaussian filter to it. The agent utilizes a convolutional layer and a fully connected layer to learn the parameters of the Gaussian filter (filter location, isotropic variance, scalar intensity) and applies this gaussian filter to the board to fixate on a specific subspace of the board (and then further processes the fixated board). We saw that the agent eventually ended up just fixating in the same place on the board throughout the episode, and so did worse than the agents reported in the manuscript (effectively just using less information than agents that take the full board). We believe this failure of the fixation model is because the effectiveness of fixation in this case depends on the agent’s action space correspondingly changing (ego-centrically with fixation). rThis is not a problem for other applications of these attention mechanisms (Mnih et al. 2014; Gregor et al. 2015) because the output space does not change with fixation, but reinforcement learning involving dynamic action spaces (rather than fixed, static action spaces) is a non-trivial engineering problem that does not have a direct solution yet. We attempt addressing this by training a multi-action version of the agent, where the agent outputs two actions: one that specifies which 3x3 square to fixate on within the 7x7 board and the other action that specifies which of those nine squares it wants to click on. Effectively, this forces the agent to learn a one-hot attention encoding on each of the 3x3 sub-boards within the 7x7 boards in the form of an action. Unfortunately, with limited time to optimize architectures and hyperparameters this agent also did not do as well as the agents in the current manuscript that look at the whole board.
> > We very much appreciate the reviewer’s insight in suggesting the crucial role of fixation, and believe it is a very important direction of future research. However, we believe that this approach (along with other possibilities like graph neural networks or neurosymbolic methods, as we discuss in the end of our discussion), is one of many ways to invoke an architectural inductive bias toward compositionality. These kinds of approaches are precisely what we are arguing for in our paper. Our goal is not to say that meta-learning compositional structure is impossible, but rather that these architectural biases are crucial when considering what meta-learners learn -- it is not sufficient to simply encode the requisite structure in the training distribution as is sometimes proposed. We also discuss this point further in response to the reviewers next point below.

---

> > > ### Author Response · Authors · 2020-11-22
> > > **We agree with this review and have significantly reframed the manuscript and briefly tested out the suggested models [3/5]**
> > >
> > > > The paper may not be able to address all the ways in which the model's experience of the task is unlike humans, but then the there should be a corresponding tempering of the conclusion that the comparison to humans says something specific about the difference between the model and humans. That is, given the current experiments, the discussion of this paper should focus at least as much on the limitations of the present model as on general conclusions about failures of the model class and the need for additional inductive biases.
> > >
> > > We agree with this reviewer’s comment and have stressed that our goal is to show a specific difference between the specific classes of models we use here,  and humans. We strongly believe that there exist model classes that have a preference for the compositional distribution -- in fact we believe the fixation model proposed by the reviewer would prefer this as well. The key point in our work is that in order to meta-learn structure implicit in training distributions, we must also consider the architecture of the meta-learner being used rather than solely considering its task distribution. Some previous work in this line of research has implied that engineering task distributions is enough to meta-learn specific inductive biases, whereas we argue that a meta-learner’s meta-inductive bias is also important to consider. We add the following changes to make this explicit:
> > > “Although the architecture we investigate here does not successfully meta-learn the ability to recognize and use abstract compositional rules, our point is not that this inductive bias cannot be meta-learned. Rather, it is that every meta-learning architecture has its own meta-inductive bias, and we show a specific case in which a standard and widely-used architecture's meta-inductive bias leads to encoding statistical features rather than the low-dimensional compositional rules used to generate the task distribution. When setting out to meta-learn a structured representation using a meta-learning system, it is important to consider the meta-inductive bias of that system in addition to engineering its task distribution. Graph neural networks \citep{battaglia2018relational}, neurosymbolic approaches \citep{ellis2020dreamcoder}, as well as attention mechanisms that allow the decomposition of the input space \cite{mnih2014recurrent}, are potential meta-learning architectures with meta-inductive biases that favor structured forms of reasoning such as compositionality.” p. 9 (Discussion)

---

> > > > ### Author Response · Authors · 2020-11-22
> > > > **We agree with this review and have significantly reframed the manuscript and briefly tested out the suggested models [4/5]**
> > > >
> > > > > Furthermore, exploring compositionality in toy tasks can be misleading. Hill et al. (2020) show that compositional generalization is significantly improved in more realistic settings (for example an RL agent that executes actions over time achieves 100% compositional generalization on a task that a feed-forward classifier only achieves 80% generalization on). They argue that toy stimuli remove one of the most important elements for training deep models — the rich environments in which humans, also, are trained. Even if the input and output of the model and humans were better matched on this dataset, it may be misleading to conclude something as general as "the importance of endowing artificial systems with this [compositional] bias" without giving these models training on a distribution of stimuli and tasks that more closely match the rich variety which humans experience over development. Of course, it is not feasible in practice to do so (yet). But this limitation and its relevance to the conclusions should at least be acknowledged in the discussion.
> > > >
> > > > We very much agree with this point by the reviewer, and have acknowledged this point in our discussion section.
> > > > “Although we encourage exploring the space of architectural changes to give meta-learning agents a better chance to learn such structured forms or reasoning, it may also be true that simpler architectures can acquire relevant inductive biases if given a rich enough high-dimensional training environment that rivals the environment(s) in which the species has evolved and individuals learn \citep{hill2019environmental}. However, the amount of data required to acquire these structured forms of reasoning in "vanilla" architectures may be prohibitively large and largely infeasible. Further, even as training within these extraordinarily large environments becomes more feasible, the biases of the correspondingly large networks being used may continue to affect the ease at which they can learn structured representations. Therefore, it is still worthwhile to investigate the role of architectural modifications on the ability to meta-learn structured representations in smaller environments, such as the ones we present here, so that one day we may transfer those insights to training on larger, more naturalistic environments. An exciting direction for future work is to examine a range of approaches to learning structured representations with the tools we set forth in this paper, and using the resulting insights to move toward closing the gap between human and machine intelligence.” p. 9 (Discussion)

---

> > > > > ### Author Response · Authors · 2020-11-22
> > > > > **] We agree with this review and have significantly reframed the manuscript and briefly tested out the suggested models [5/5]**
> > > > >
> > > > > > The paper could use some more discussion of the distinction between statistical patterns and compositional rules. This seems like an important point, but it wasn't entirely clear to me. Compositional rules correspond to a certain statistical distribution over grids. The fact that up to 2nd order Ising statistics are matched does not mean that the distributions of outcomes are matched "statistically," it merely means they match in certain low-order statistics. It's not clear if these are the right statistics by which to compare the distributions (especially for the non-convolutional model, which has no spatial awareness). The paper would be strengthened by justifying the choice of these statistics more carefully, and articulating a clear distinction between what counts as a "statistical" pattern vs. a rule.
> > > > >
> > > > > We acknowledge the lack of clarity of these technical terms and thank the reviewer for an opportunity to clarify. We briefly summarize our clarifications here and refer to relevant changes in the manuscript. Compositional structure, for our purposes, is the property of a few low-dimensional, simple rules that can be composed together to generate arbitrary complexity. The term statistical structure, for our purposes, refers to the global statistics as reflected in the conditional distributions of red and blue tiles across the whole board. These statistics can emerge from the compositional boards produced from a generative grammar, but can also be closely matched without the explicit use of low dimensional rules (as reflected in our null distribution).
> > > > >
> > > > > We would also like to point out that to generate the null distribution, we trained a network to almost perfect accuracy on the conditional distributions at the full-board level that emerge from the compositional distribution. We do not use Ising statistics as the de-facto statistic to show that these two distributions are matched, we simply use them as an intuitive sanity check.
> > > > >
> > > > > Finally, we also highlight that it is possible that the conditional distributions do not match perfectly, making one “harder” or “easier” than the other. But the key finding is that which is harder or easier is different between humans and machines (across both the feedforward and compositional agents). This highlights a crucial difference between humans and machines, and we argue that which distribution they find easier to learn reflects differences in their inductive biases. (humans are better at for abstract, structured rules whereas the agent prefers  statistical patterns).
> > > > >
> > > > > Relevant change to the manuscript:
> > > > > “This control distribution allows us to disentangle statistical pattern matching from structured reasoning (which, in this specific case, is rule-based compositionality) and highlights the difference between actually learning and utilizing simple abstract structures (e.g. low-dimensional compositional rules) versus using the statistical patterns that may be a downstream consequence of those structures. Our method closely approximates the global statistics that emerge from the compositional rules, by using a neural network to learn the conditional distributions and generating Gibbs samples from these conditionals.” p. 8-9 (Discussion)

---

> > ### Comment · AnonReviewer3 · 2020-11-23
> > **Thanks for the revision, I think the paper has improved substantially, but I still have some lingering concerns.**
> >
> > First, I want to acknowledge the substantial revision of the paper, which I think has greatly improved it. I also appreciate them running experiments using a fixation model, although I understand these models can be difficult to train correctly. I will revise my rating of the paper upward accordingly.
> >
> > However, I still have some lingering concerns. Since the discussion period is nearly over, it may be too late to address them here, but perhaps they will be useful for the final version.
> >
> > 1) I appreciate the analyses of the different human performance on the different task types. However, more importantly I think there are still a few more comparisons which would be very illuminating vis-a-vis the framing of the paper. In particular, since humans do worse at the tree structure tasks than chains and loops, are they still performing better at the tree tasks than the non-compositional tasks? Are they performing better at tree tasks than the (convolutional) model performs at the tree tasks? Again, I think that these comparisons are important to the claim that participants are actually better at abstract compositional rules, rather than some other feature. While you say: "We argue that even the 'simplicity' of a chain is due to the simplicity of the underlying rule that generates it (to recursively extend in opposite directions)," it's not clear to me that this is the right way to think about the simplicity. My introspection (admittedly biased) is that if I were performing the tasks I would see chains as straight lines, rather than thinking about them in terms of recursive rules that generate them. If humans are not outperforming the models at trees, should that alter our conclusions about whether abstract compositional rules are actually the underlying factor differentiating performance on these two distributions? I feel like the lack of clarity around these issues is an obstacle to considering this a strong paper.
> >
> > Perhaps the easier takeaway from this point (regardless of the answer to the above question) is that I think the supplement would benefit from a detailed breakdown of human and both model's performance (in a straight-forward metric like blue tiles revealed) for every condition: chains, loops, trees, and noncompositional. That might help your readers to generate more specific hypotheses about what's missing from the models and how it could be incorporated.
> >
> > 2) I'm more confused about what you would consider a compositional inductive bias than I was before the discussion. You say "in fact we think that allowing for fixation might precisely be the compositional inductive bias required here." What makes this a compositional inductive bias? It certainly doesn't directly allow for the abstraction of rules, although it does allow for abstraction over positions, which is perhaps what you mean. I think the paper would still benefit from a more detailed discussion of what counts as a compositional bias.

---

> > > ### Author Response · Authors · 2020-11-24
> > > **We thank the reviewer for their reevaluation of our manuscript, as well as for their quick response and follow up comments. We have revised our paper accordingly [1/2]**
> > >
> > > > 1) I appreciate the analyses of the different human performance on the different task types. However, more importantly I think there are still a few more comparisons which would be very illuminating vis-a-vis the framing of the paper. In particular, since humans do worse at the tree structure tasks than chains and loops, are they still performing better at the tree tasks than the non-compositional tasks? Are they performing better at tree tasks than the (convolutional) model performs at the tree tasks? Again, I think that these comparisons are important to the claim that participants are actually better at abstract compositional rules, rather than some other feature. While you say: "We argue that even the 'simplicity' of a chain is due to the simplicity of the underlying rule that generates it (to recursively extend in opposite directions)," it's not clear to me that this is the right way to think about the simplicity. My introspection (admittedly biased) is that if I were performing the tasks I would see chains as straight lines, rather than thinking about them in terms of recursive rules that generate them. If humans are not outperforming the models at trees, should that alter our conclusions about whether abstract compositional rules are actually the underlying factor differentiating performance on these two distributions? I feel like the lack of clarity around these issues is an obstacle to considering this a strong paper.
> > > Perhaps the easier takeaway from this point (regardless of the answer to the above question) is that I think the supplement would benefit from a detailed breakdown of human and both model's performance (in a straight-forward metric like blue tiles revealed) for every condition: chains, loops, trees, and noncompositional. That might help your readers to generate more specific hypotheses about what's missing from the models and how it could be incorporated.
> > >
> > > This is a great suggestion by the reviewer and we have included these additional analyses in the appendix section A. 4 (Figure 8). Comparisons across human/agent/conv agent in the chain/tree/loop/null conditions are shown. Indeed, despite the humans struggling most at the tree tasks of all the compositional tasks, they still perform significantly better at tree boards than at the null boards. This is consistent with the claim that humans utilize low-dimensional rules rather than emergent statistical structure, and that the “straight line” simplicity of the chain boards is not driving the effects.
> > > We do find that the humans do worse than the convolutional agent on tree structures. This is possibly due to a combination of tree-specific poor human performance (it is the most complex rule of the three types and humans consistently perform worst at it, though, crucially, still better than at the null tasks) and by tree-specific good agent-performance (perhaps the statistical properties of trees are best suited to the agent, note that performance of both agents on trees is very close to their performance on the non-compositional null distribution). Further analyses of these are valuable directions of future research.
> > > We note another observation that we believe furthers the claim that the key difference between humans and agents is the use of low-dimensional rules: that humans vastly outperform both agents in the loop condition. The loop production rule is quite simple, but cannot trivially be described as a “straight line”.

---

> > > > ### Author Response · Authors · 2020-11-24
> > > > **We thank the reviewer for their reevaluation of our manuscript, as well as for their quick response and follow up comments. We have revised our paper accordingly [2/2]**
> > > >
> > > > > 2) I'm more confused about what you would consider a compositional inductive bias than I was before the discussion. You say "in fact we think that allowing for fixation might precisely be the compositional inductive bias required here." What makes this a compositional inductive bias? It certainly doesn't directly allow for the abstraction of rules, although it does allow for abstraction over positions, which is perhaps what you mean. I think the paper would still benefit from a more detailed discussion of what counts as a compositional bias.
> > > >
> > > > We thank the reviewer for an opportunity to clarify. Precisely as the reviewer suggests, fixation permits abstraction over positions. In other words, it implicitly encourages the system to learn a representation that consists of a pattern for each fixation point that can be applied across different fixation points. This happens to be exactly the compositional structure that underlies our generative process. We start with a simple pattern (the production rules as defined by the nonterminal symbols in Figure 1) at each “fixation point” (the starting point for the production), and generate a macroscopically complex pattern by repeating this over different fixation points. Explicitly encoding fixation thereby gives the system an inductive bias toward the specific compositional structure in our task distribution. However, we completely agree that this does not present a bias toward rules per se. Further, fixation does not favor compositionality in its most general form, only the kinds of local spatial compositionality used in our distribution. In the interest of keeping things more general, we edit the discussion to argue more broadly for inductive biases toward structured representations:
> > > >
> > > > “Graph neural networks (Battaglia et al., 2018), neurosymbolic approaches (Ellis et al., 2020), as well as attention mechanisms (Mnih et al., 2014), permit abstraction by (implicitly or explicitly) decomposing the input into parts. Using these in meta-learning architectures might favor structured representations and reasoning. ”

---

> > > > > ### Comment · AnonReviewer3 · 2020-11-24
> > > > > **Thanks, this is helpful!**
> > > > >
> > > > > Thanks again to the authors for being very fast and responsive! I really like the presentation of the results in the appendix, it shows some very interesting patterns that I think will provide a lot of fuel for future research. I will keep this in mind during the discussion phase!

---

### Official Review · AnonReviewer4 · 2020-10-21
**Very interesting work**

**Rating:** 7
**Confidence:** 3

**Review:**

*Summarize what the paper claims to contribute. Be positive and generous.*

This paper offers an interesting comparison between humans and a meta-learning algorithm [1,2] learning to uncover structured patterns on a 7x7 grid. The patterns are either generated using simple “compositional rules” (lines, loops or trees) or sampled from a distribution that has almost identical 0th, 1st and 2nd order statistics as the ones generated through the “compositional rules”. The board is initially covered, and the agent (either a human or a RL meta-learning algorithm) is tasked with selecting one by one which tiles are to be uncovered. The aim is to guess which tiles are covered by the pattern (red tiles) and avoid the background (blue tiles), thus uncovering the red tiles as accurately as possible. The authors go on to show supporting evidence that most likely, humans are using a compositional inductive bias when trying to uncover the hidden pattern, whereas the meta-learning algorithms do not. This conclusion is arrived at using the relative performance of both humans and the algorithm on the cases where the pattern was generated using the two different schemes. They then go on to conclude that there is a strong difference between the strategy learned by the algorithm, which seems to use mostly the statistics, compared to humans that seem to make use of a compositional inductive bias.


[1] Yan Duan, John Schulman, Xi Chen, Peter L Bartlett, Ilya Sutskever, and Pieter Abbeel. Rl^2: Fast reinforcement learning via slow reinforcement learning. arXiv preprint arXiv:1611.02779, 2016.
[2] Jane X Wang, Zeb Kurth-Nelson, Dhruva Tirumala, Hubert Soyer, Joel Z Leibo, Remi Munos, Charles Blundell, Dharshan Kumaran, and Matt Botvinick. Learning to reinforcement learn. arXiv preprint arXiv:1611.05763, 2016.


*List strong and weak points of the paper. Be as comprehensive as possible.*

**Strong points**

- The paper is very clear and very well written. I like the simplicity of the experiment and the approach taken by the authors in trying to uncover whether humans and a particular algorithm follow similar strategies when solving tasks.
- I see no reason why this approach should not be re-used by other researchers, and I hope that it will inspire others to go to similar lengths when analysing the operation of their algorithms. Specifically, I consider this a strong point of the paper as I believe it offers an obvious and accessible set of future work.

**Weak points**

- It is not completely clear how the statistically similar patterns are constructed. Is it possible to get patterns that are identical to the compositionally generated ones? If so, have you checked and removed these? Please elaborate a bit more on this aspect, show more examples of the non-compositional patterns, and describe how you decided on the Compositional-passing/not compositional passing split in Figure 3C.
- One somewhat deeper weakness of this work is that one can claim that there is no inherent notion of compositionality in the patterns of the data per-se, but rather in the manner through which they were generated. This is perhaps an underlying reason why the meta-learning algorithm never picks up the compositionality inductive bias exhibited by humans. My point does not mean to invalidate the findings and conclusion, but rather to emphasise that perhaps we shouldn’t be looking at this type of experiments, as they are “doomed to fail”. Of course this might be obvious in hindsight, but I would be interested to read what the authors think about this.


*Clearly state your recommendation (accept or reject) with one or two key reasons for this choice. Provide supporting arguments for your recommendation.*

I recommend accepting the paper, mainly for the two strong points earlier. I believe what the length at which authors have gone to for understanding what an algorithm does (or what it doesn’t do) should be an example for our field, which often lacks imagination when it comes to analysing proposed models. I do think that a condition for accepting is at least ensuring that the details of all the generated samples (compositional or not) are included in the paper. How many (unique) of each were generated/used? How did you decide the compositional-passing/not compositional passing split. Did you check whether any non-compositional pattern matched the compositional one? If so, how and did you reject that sample?

*Provide additional feedback with the aim to improve the paper. Make it clear that these points are here to help, and not necessarily part of your decision assessment.*

- In the intro: “Second, humans represent this learned information compositionally”. In my opinion this is a very strong statement about the nature of the representation in the human brain and it’s best avoided unless there’s neurophysiological evidence. Consider rephrasing to a softer version with a reference.
- 4th paragraph in the introduction: “Our methodological contribution in this work is *to* develop novel tasks.”
- Last paragraph of introduction: “Since a large swath of real-world tasks contain compositional structure..” Which ones? Please include some examples. Also, is it that they contain compositional structure (i.e. in the way they are generated) or that the compositional inductive bias that humans exhibit can more efficiently solve them?
- Second sentence of 3.2 Results: “We demonstrate that humans have a clear bias toward compositional distributions, without extensive training and even while directly controlling for statistical complexity.” Couldn’t we claim that the patterns generated by the compositional distributions are easier/simpler? They are after all only 3 different rules. Wouldn’t that be a good enough explanation of why humans solve those much more easily than the statistically matched ones? You did touch upon the issue of spatial proximity later on. Can you touch upon this point too? I think it’s a possible (and perhaps simpler) alternative explanation of why humans would be better at them.
- Appendix A.1. The hyperparameters are reported in an unnecessary accuracy - I would assume that results are not sensitive to that many significant figures. Please consider 2 or 3 s.f. in scientific notation for simplicity.

---

> ### Author Response · Authors · 2020-11-22
> **We appreciate the positive review. We have tried to improve the clarity of the manuscript.**
>
> We thank the reviewer for overall positive comments. Below is a point-by-point response with the relevant changes to the manuscript:
> > It is not completely clear how the statistically similar patterns are constructed. Is it possible to get patterns that are identical to the compositionally generated ones? If so, have you checked and removed these? Please elaborate a bit more on this aspect, show more examples of the non-compositional patterns, and describe how you decided on the Compositional-passing/not compositional passing split in Figure 3C.
>
> We apologize for the ambiguity in this analysis. It is indeed possible to get some boards identical to the compositionally generated one. While there is overlap due to the overall distributions being statistically similar, the null distribution is not underlied by a small set of simple rules (as opposed to the compositional distribution). The following change to the manuscript describes the compositional-passing/not compositional-passing to greater detail:
>
> “We split the null test set by whether or not the board is `compositional-passing' and compare human performance across these. To do this, we generated the set of all possible compositional boards on a $7\times 7$ grid and labeled any null task distribution board as compositional-passing if it happened to be a part of this set.” p. 6 (Results)
>
> > One somewhat deeper weakness of this work is that one can claim that there is no inherent notion of compositionality in the patterns of the data per-se, but rather in the manner through which they were generated. This is perhaps an underlying reason why the meta-learning algorithm never picks up the compositionality inductive bias exhibited by humans. My point does not mean to invalidate the findings and conclusion, but rather to emphasise that perhaps we shouldn’t be looking at this type of experiments, as they are “doomed to fail”. Of course this might be obvious in hindsight, but I would be interested to read what the authors think about this.
>
> We agree that the manner in which the boards were generated across the task distributions is where the fundamental difference lies. Since the task involves uncovering the red tiles, an understanding of how the boards are generated would be very useful to perform well. Figures 4 and 5B speak to the specific process that humans use to perform this task and it is consistent with having encoded simple generative rules. By contrast, the Gibbs sampling procedure generates null distribution boards based on the global spatial statistics of the compositional boards through a process that does not involve the use of specific simple composable rules. We see differences in human and agent performance on these two
>
> > In the intro: “Second, humans represent this learned information compositionally”. In my opinion this is a very strong statement about the nature of the representation in the human brain and it’s best avoided unless there’s neurophysiological evidence. Consider rephrasing to a softer version with a reference.
>
> We agree and have rephrased this claim.
>
> > Second sentence of 3.2 Results: “We demonstrate that humans have a clear bias toward compositional distributions, without extensive training and even while directly controlling for statistical complexity.” Couldn’t we claim that the patterns generated by the compositional distributions are easier/simpler? They are after all only 3 different rules. Wouldn’t that be a good enough explanation of why humans solve those much more easily than the statistically matched ones? You did touch upon the issue of spatial proximity later on. Can you touch upon this point too? I think it’s a possible (and perhaps simpler) alternative explanation of why humans would be better at them.
>
> The reviewer is absolutely right, and we make changes to the manuscript to stress this point. We believe that it is the fact that humans are biased towards learning low-dimensional rules (that can be composed to form abstract structures) that makes these compositional boards “easier” and leads them to do better. This is as opposed to the null distribution that has similar global statistical properties as those that emerge from the composition of low-dimensional rules, but are not explicitly generated using these simple rules. We have emphasized this point in our revised manuscript.

---

> > ### Comment · AnonReviewer4 · 2020-11-23
> > **Thank authors for response**
> >
> > I would like to thank the authors for their detailed response to my review. I am very satisfied with all their answers, which answer all my questions and concerns.

---

### Official Review · AnonReviewer2 · 2020-10-28
**An interesting exploration that could have benefited from a more precise definition of terms**

**Rating:** 6
**Confidence:** 3

**Review:**

This paper sets out to determine something about the form of the bias acquired by a standard meta-learning algorithm, and compare the form of that bias to the inherent bias that humans have.    The authors point out, rightly, that meta-learning algorithms have meta-biases and it is important to understand these, from both the scientific and engineering perspectives.  It is well written and raises good questions.

There is a cleverly constructed test domain and a set of well-executed computer and human experiments (I think---I don't really know about how to construct a human experiment.)

Unfortunately, I can't end up agreeing or disagreeing with the claims made in the paper, or really understands how well they are supported by evidence, because I find that they use terms that don't seem to be sufficiently technically well defined.

For example:
- what exactly is compositional structure?
- what is statistical structure?
- what is your measure of task complexity?

How can we tell if what the agent learns is compositional?  Is that an externally measurable property of the agent's behavior and the way it generalizes to new environments?  Or is it a property of the internal representation?   (It is common to have an intuition that "compositional" also implies "compact" or "low complexity" in some sense.)

It feels like generalization be a way to get more clearly at the presence of a compositional representation:  could you train on small grids and have the learned agent generalize to big ones?  It seems like if a fixed-size representation can generalize to very large instances, then that is more clear evidence of compositionality (but then I'm thinking of compositionality as a property of a representation, not of externally-measurable behavior.)

I also feel that I don't quite understand the meta-learning training regime.  What exactly constituted a "task" from the meta-learning perspective?   Is it a single board?  If so, then the meta-learning problem is to learn the task distribution, in some sense.   I was expecting something more "meta":  that is, to test whether the system is actually meta-learning the *idea* of compositionality, it seems like set-up would be that a task corresponds to a particular grammar with multiple boards drawn from the distribution induced by the grammar;  then we'd know that it had meta-learned compositionality if it could learn *new grammars* quickly.

Smaller points
- I didn't completely understand the production rule (nor the examples) for the loop structure.
- It would help me understand the task set better if there were a slightly more in-depth description of the chains, trees, and loops and described how the grammar generates the compositional tasks in figure 2. Is it not the case that every connected configuration of red tiles could be described as a tree?
- Rather than showing just one number for the final performance, It would be helpful to show learning curves for the RL algorithms so the reader can assess the stability, convergence, etc. Similarly, learning curves for humans would be interesting, but less important since I assume they just look flat.
- "is develop"

---

> ### Author Response · Authors · 2020-11-22
> **We have rewritten the manuscript to significantly improve the clarity [1/2]**
>
> We thank the reviewer for their helpful comments. Below is our point-by-point by response to their comments along with relevant manuscript changes.
>
> > This paper sets out to determine something about the form of the bias acquired by a standard meta-learning algorithm, and compare the form of that bias to the inherent bias that humans have. The authors point out, rightly, that meta-learning algorithms have meta-biases and it is important to understand these, from both the scientific and engineering perspectives. It is well written and raises good questions.
> There is a cleverly constructed test domain and a set of well-executed computer and human experiments (I think---I don't really know about how to construct a human experiment.)
> Unfortunately, I can't end up agreeing or disagreeing with the claims made in the paper, or really understands how well they are supported by evidence, because I find that they use terms that don't seem to be sufficiently technically well defined.
> > For example:
> > * what exactly is compositional structure?
> > * what is statistical structure?
> > * what is your measure of task complexity?
> > How can we tell if what the agent learns is compositional? Is that an externally measurable property of the agent's behavior and the way it generalizes to new environments? Or is it a property of the internal representation? (It is common to have an intuition that "compositional" also implies "compact" or "low complexity" in some sense.)
>
> We acknowledge the lack of clarity of these technical terms and thank the reviewer for an opportunity to clarify. We briefly summarize our clarifications here and refer to relevant changes in the manuscript. We have reframed the paper significantly to focus more broadly on meta-learning as an approach to learning structured forms of reasoning, and present compositionality as one such example of structure. Compositional structure, for our purposes, is the property of a few low-dimensional, simple rules that can be composed together to generate arbitrary complexity (as in a grammar). The intuition that the reviewer has of it being compact and low-complexity is correct.
> The term statistical structure, for our purposes, refers to the global statistics as reflected in the conditional distributions of red and blue tiles across the whole board. These statistics can emerge from the compositional boards produced from the generative grammar in Figure 1, but can also be closely matched without the explicit use of compositional rules (as reflected in our null distribution). Our insight is that humans can recognize these abstract structures and thus do better in the compositional task distribution whereas the preferences is flipped for agents, which prefer the use of global statistics and thus do better in the null task distribution.
> Finally, we do not directly measure task complexity in this work. Although the two task distributions have similar global statistics, it could very well be that one task distribution is simply more complex than the other. However,the core of our results lies in the double dissociation of performance in human vs agents (humans do better on compositional, agents do better on null). In fact a key point in our paper is that “task complexity” is not an objective measure since humans and agents find different tasks difficult, depending on their biases.
>
> The following are related changes to the manuscript:
>
> “This control distribution allows us to disentangle statistical pattern matching from structured reasoning (which, in this specific case, is rule-based compositionality) and highlights the difference between actually learning and utilizing simple abstract structures (e.g. low-dimensional compositional rules) versus using the statistical patterns that may be a downstream consequence of those structures. Our method closely approximates the global statistics that emerge from the compositional rules, by using a neural network to learn the conditional distributions and generating Gibbs samples from these conditionals.” p. 8-9 (discussion)

---

> > ### Author Response · Authors · 2020-11-22
> > **We have rewritten the manuscript to significantly improve the clarity [2/2]**
> >
> > > It feels like generalization be a way to get more clearly at the presence of a compositional representation: could you train on small grids and have the learned agent generalize to big ones? It seems like if a fixed-size representation can generalize to very large instances, then that is more clear evidence of compositionality (but then I'm thinking of compositionality as a property of a representation, not of externally-measurable behavior.)
> >
> > We agree with the reviewer that this is a good way to measure compositional representations. However, we believe that this has been done before in much of the previous literature. Part of the novelty of our work is to consider how the recognition of abstract structure results in externally-measurable behavior. Figures 4 and 5B captures this, where human behavior is qualitatively more consistent with recognizing abstract structure whereas the agent acts in a way that suggests that they are relying more on global statistics. We believe these analysis tools expand the scope of how we can better understand the procedures learned by agents and compare them with humans.
> >
> > > I also feel that I don't quite understand the meta-learning training regime. What exactly constituted a "task" from the meta-learning perspective? Is it a single board? If so, then the meta-learning problem is to learn the task distribution, in some sense. I was expecting something more "meta": that is, to test whether the system is actually meta-learning the idea of compositionality, it seems like set-up would be that a task corresponds to a particular grammar with multiple boards drawn from the distribution induced by the grammar; then we'd know that it had meta-learned compositionality if it could learn new grammars quickly.
> >
> > In our paper, one task is a specific board and a distribution of tasks over which we meta-learn is to solve boards from a generative process. This generative process involves composing simple low-dimensional rules to generate complex shapes. It is easier to perform well on a new board if these composable rules have been learned. We agree that the natural next step is to additionally meta-learn over different possible rules / generative processes. This is precisely the kind of hierarchical structure that is well supported by such compositional grammars and we are excited about future work in this direction. In this paper we demonstrate interesting differences between humans and agents even in this relatively simple task, without having to meta-learn over the space of possible grammars.
> >
> > > * I didn't completely understand the production rule (nor the examples) for the loop structure.
> > > * It would help me understand the task set better if there were a slightly more in-depth description of the chains, trees, and loops and described how the grammar generates the compositional tasks in figure 2. Is it not the case that every connected configuration of red tiles could be described as a tree?
> > > * Rather than showing just one number for the final performance, It would be helpful to show learning curves for the RL algorithms so the reader can assess the stability, convergence, etc. Similarly, learning curves for humans would be interesting, but less important since I assume they just look flat.
> > > * "is develop"
> >
> > We thank the reviewer for these helpful comments. The information requested here (lengthier description of generative grammar and training curves) has been added to the appendix.

---

### Official Review · AnonReviewer1 · 2020-10-30
**Some nice work, but contains overly broad claims and interpretations**

**Rating:** 6
**Confidence:** 4

**Review:**

This work is an exploration of model behaviour upon meta-learning tasks with compositional structure. The authors discover that, unlike humans, machine learning models do not readily pick up on the underlying compositional generative structure of a set of tasks, and hence cannot match the performance of humans. Conversely, when the task is structured to leverage other statistical patterns, models do well.

Taken as a whole, this is a nice piece of work. The presentation is well crafted, and I believe the experiments are well planned. There are many nice analyses and some welcome statistics, such as shown in Figure 3. The authors are commended for their work. I wish to lay out a few criticisms, and I'd like the authors to know that the points are all very easily fixable.

The authors design a set of structure-learning tasks using a generative grammar. The exact details of the grammar are not given, and the reader is to rely on rough intuitions based on the figures. I encourage the authors to spell out some more details of the methods.

The authors argue that the non-compositional boards could not have been generated by the defined grammar, which seems fair, but they also argue that these boards are necessarily non-compositional. I am having a lot of trouble with this statement, because it is not entirely clear by what the authors mean by compositional, as it hasn't been clearly defined. This is a particular problem in the machine learning field as a whole, as it pertains to research on compositionality; rarely is the term defined in any rigorous sense, and from paper to paper there are seemingly different definitions. I encourage the authors to clearly explain what entails compositionality as they refer to it here, and to explain what makes their non-compositional boards non-compositional according to their definition. The "non-compositional" boards might not match the generative grammar as used in the compositional setting, but it does not entail that there does not exist a compositional grammar that can produce the non-compositional boards. In fact, due to the discrete, simplified nature of the game, it would seem almost certainly true that there exists *some* generative grammar that can produce the non-compositional boards seen here. It might not be a "simple", "interesting", or human interpretable one, but it would nonetheless be a grammar, and would be compositional. This fact makes it all the more important to define compositionality.

It seems to me that what is more precisely being illustrated is the ability for humans to perceive, and infer the implications of abstract, "simple" *structures*, and not compositional rules or grammars per se. Without fully defining compositionality and establishing the non-compositionality in the null setting, I'm afraid that the results are misstated and misrepresented. A note to the authors: I'm fully aware that I could be misunderstanding how the MLP+Gibbs sampling method here can entail non-compositionality, and am more than open to being corrected on this matter. I look forward to a discussion in the rebuttal.

The previous point leads me to a broader point about the background presented throughout. The authors include many broad, and quite bold statements regarding humans, and how they learn, especially in regards to their capacity for "learning compositionality". It is claimed that humans learn rapidly, with very few samples, which is contrasted with machine models that require an enormous amount of data. This is true in some superficial sense, but does not account for the the entire evolutionary trajectory that produced humans; indeed, humans at birth are not blank slates to the degree that randomly initialized neural networks are. It's also claimed that humans learn compositional representations. While this idea is certainly in vogue in some circles in cognitive science, it is certainly not widely agreed upon. I'm not even sure how such a strong statement can be proven. The citations given point to a cople computational modeling papers, which, in my opinion, insufficiently corroborate such a claim. Moreover, it's not even clear how many non-artificial pieces of data that humans deal with are even truly compositional (for a taste of the issues and controversy surrounding one particular example --- language --- see the following entry: https://plato.stanford.edu/entries/compositionality/). Please note that this is not at all to say the views presented in this paper are necessarily false. I merely suggest that the authors take another pass at their writing, and tame a few of the broader, bolder claims about the nature of human learning, because it's not clear that they are necessarily true, either.

I believe the authors have missed out on some possible interpretations to their results. They claim that a meta-learned model cannot learn the compositional structure of the tasks, since meta-learning is insufficient to establish the inductive biases required for compositional understanding. However, the inductive bias of a model is determined by more than its parameters (which in this case, are established via meta-learning). The functions a model comes to learn are also dependent on the nature of the computations, manifest through the architecture, and other such things. The authors are well aware of this, since they include a condition wherein the model uses convolutions. But the fact that the convolutional model does better entails that there might exist further architectural variants that do even better than it, and potentially, better than humans. If this were the case, then we'd no longer be able to claim that meta-learning cannot establish the proper inductive bias for compositional understanding. Therefore, the existence of a gradient in model performances warrants more careful wording in regards to the claims; we cannot so broadly categorize meta-learning as insufficient for establishing the right inductive biases for compositional understading without caveating according to the other sources of inductive bias. One piece of proof that the humans understand the compositional structure of the task is that they improve over the course of the task. But could this not simply reflect the fact that humans have a better capacity to improve behaviour over short time periods? In other words, how do we know that the problem with the models doesn't have to do with, say, working memory, rather than compositional understanding per se?

Altogether, this is a well put together paper. There is a lot of interesting work here, and the authors have done well to explore various facets of the setup. My main criticisms have to do with the way the work is pitched, and the way the results are interpreted. It is very much presented with a veneer that speaks to a very particular crowd in cognitive-science inspired machine learning community. But since the views in this community are not necessarily broadly shared, many of the statements, interpretations, and claims come across as quite strong and not fully corroborated.

---

> ### Author Response · Authors · 2020-11-22
> **We have significantly reframed the paper in accordance to these comments [1/5]**
>
> We thank the reviewer for their helpful comments and have made a response for each of their comments along with suggested changes to the paper:
>
> > This work is an exploration of model behaviour upon meta-learning tasks with compositional structure. The authors discover that, unlike humans, machine learning models do not readily pick up on the underlying compositional generative structure of a set of tasks, and hence cannot match the performance of humans. Conversely, when the task is structured to leverage other statistical patterns, models do well. Taken as a whole, this is a nice piece of work. The presentation is well crafted, and I believe the experiments are well planned. There are many nice analyses and some welcome statistics, such as shown in Figure 3. The authors are commended for their work. I wish to lay out a few criticisms, and I'd like the authors to know that the points are all very easily fixable.
>
> We thank the reviewer for their nice summary and commendation of our work.
>
> > The authors design a set of structure-learning tasks using a generative grammar. The exact details of the grammar are not given, and the reader is to rely on rough intuitions based on the figures. I encourage the authors to spell out some more details of the methods.
>
> We agree our initial description of our generative grammar was too vague. We have included a more detailed description in the appendix.

---

> > ### Author Response · Authors · 2020-11-22
> > **We have significantly reframed the paper in accordance to these comments [2/5]**
> >
> > > The authors argue that the non-compositional boards could not have been generated by the defined grammar, which seems fair, but they also argue that these boards are necessarily non-compositional. I am having a lot of trouble with this statement, because it is not entirely clear by what the authors mean by compositional, as it hasn't been clearly defined. This is a particular problem in the machine learning field as a whole, as it pertains to research on compositionality; rarely is the term defined in any rigorous sense, and from paper to paper there are seemingly different definitions. I encourage the authors to clearly explain what entails compositionality as they refer to it here, and to explain what makes their non-compositional boards non-compositional according to their definition. The "non-compositional" boards might not match the generative grammar as used in the compositional setting, but it does not entail that there does not exist a compositional grammar that can produce the non-compositional boards. In fact, due to the discrete, simplified nature of the game, it would seem almost certainly true that there exists some generative grammar that can produce the non-compositional boards seen here. It might not be a "simple", "interesting", or human interpretable one, but it would nonetheless be a grammar, and would be compositional. This fact makes it all the more important to define compositionality.
> >
> > > It seems to me that what is more precisely being illustrated is the ability for humans to perceive, and infer the implications of abstract, "simple" structures, and not compositional rules or grammars per se. Without fully defining compositionality and establishing the non-compositionality in the null setting, I'm afraid that the results are misstated and misrepresented. A note to the authors: I'm fully aware that I could be misunderstanding how the MLP+Gibbs sampling method here can entail non-compositionality, and am more than open to being corrected on this matter. I look forward to a discussion in the rebuttal.
> >
> > We agree with the reviewer’s concerns and have accordingly reframed our discussion. The boards we refer to as “compositional” have now been clarified to mean boards generated with low-dimensional composable rules. We agree that containing “simple” structures is the key aspect we are interested in and these simple structures can in turn be composed to create more complex shapes. Our principal point about the difference between these compositional boards and the null distribution boards (generated through the Gibbs sampling procedure) is that while the null distribution boards respect the conditional distributions (at the full board level) of the “compositional” boards, they do not strictly use simple, discrete rules. While specific null distribution boards could have been generated from grammar rules, the same low-dimensional rules do not underlie the whole distribution (unlike in the compositional distribution).
> > We realize that this point did not come across well in our original framing, so we have significantly rewritten the paper to emphasize this point, including changing the title of the paper. Below are some representative examples. We welcome any other suggestions for further changes to reflect this broader point:
> >
> > “In recent years, meta-learning, in which a model is trained on a family of tasks (i.e. a task distribution), has emerged as an approach to training neural networks to perform tasks that were previously assumed to require structured representations, making strides toward closing the gap between humans and machines. However, we argue that evaluating meta-learning remains a challenge, and can miss whether meta-learning actually uses the structure embedded within the tasks. These meta-learners might therefore still be significantly different from humans learners. To demonstrate this difference, we first define a new meta-reinforcement learning task in which a structured task distribution is generated using a compositional grammar.” p. 1 (abstract)
> >
> > “In this work, we make several methodological and scientific contributions to provide a rigorous way to test for structured forms of reasoning using compositional grammars as a case study. We show that human behavior is consistent with learning and utilizing low-dimensional compositional rules. We also show that standard meta-learning approaches, in sharp contrast to humans, struggle with discrete abstract structures and prefer statistical patterns.” p. 8 (discussion)

---

> > > ### Author Response · Authors · 2020-11-22
> > > **We have significantly reframed the paper in accordance to these comments [3/5]**
> > >
> > > > The previous point leads me to a broader point about the background presented throughout. The authors include many broad, and quite bold statements regarding humans, and how they learn, especially in regards to their capacity for "learning compositionality". It is claimed that humans learn rapidly, with very few samples, which is contrasted with machine models that require an enormous amount of data. This is true in some superficial sense, but does not account for the the entire evolutionary trajectory that produced humans; indeed, humans at birth are not blank slates to the degree that randomly initialized neural networks are. It's also claimed that humans learn compositional representations. While this idea is certainly in vogue in some circles in cognitive science, it is certainly not widely agreed upon. I'm not even sure how such a strong statement can be proven. The citations given point to a cople computational modeling papers, which, in my opinion, insufficiently corroborate such a claim. Moreover, it's not even clear how many non-artificial pieces of data that humans deal with are even truly compositional (for a taste of the issues and controversy surrounding one particular example --- language --- see the following entry: https://plato.stanford.edu/entries/compositionality/). Please note that this is not at all to say the views presented in this paper are necessarily false. I merely suggest that the authors take another pass at their writing, and tame a few of the broader, bolder claims about the nature of human learning, because it's not clear that they are necessarily true, either.
> > >
> > > We agree that our claims about the nature of human learning are rather broad and detract from the main point we want to make -- that humans are better at learning certain simple, low-dimensional composable rules as compared to higher dimensional statistical properties, while the agent in our work has the opposite inductive bias. We have removed the discussion of sample complexity in favor of a more general discussion of meta-learning as an approach to learning structured forms of reasoning.  Finally, we agree that we do not have enough evidence at this time to make the claim that all human representations are truly compositional and have removed this claim.

---

> > > > ### Author Response · Authors · 2020-11-22
> > > > **We have significantly reframed the paper in accordance to these comments [4/5]**
> > > >
> > > > > I believe the authors have missed out on some possible interpretations to their results. They claim that a meta-learned model cannot learn the compositional structure of the tasks, since meta-learning is insufficient to establish the inductive biases required for compositional understanding. However, the inductive bias of a model is determined by more than its parameters (which in this case, are established via meta-learning). The functions a model comes to learn are also dependent on the nature of the computations, manifest through the architecture, and other such things. The authors are well aware of this, since they include a condition wherein the model uses convolutions. But the fact that the convolutional model does better entails that there might exist further architectural variants that do even better than it, and potentially, better than humans. If this were the case, then we'd no longer be able to claim that meta-learning cannot establish the proper inductive bias for compositional understanding. Therefore, the existence of a gradient in model performances warrants more careful wording in regards to the claims; we cannot so broadly categorize meta-learning as insufficient for establishing the right inductive biases for compositional understading without caveating according to the other sources of inductive bias.
> > > >
> > > > We very much agree that meta-learning algorithms could well have compositional inductive bias and supersede human performance on compositional tasks. Exactly as suggested by the reviewer the, we believe that architectural biases will be essential to this rather than solely considering its task distribution. In this paper, we argue against the idea that simply training neural networks on large enough structured datasets is enough to meta-learn that structure -- a claim we see in the literature (around theory of mind, causality, and other examples we highlight in the introduction). We thank the reviewer for the opportunity to clarify this further.
> > > >
> > > > “Although the architecture we investigate here does not successfully meta-learn the ability to recognize and use abstract compositional rules, our point is not that this inductive bias cannot be meta-learned. Rather, it is that every meta-learning architecture has its own meta-inductive bias, and we show a specific case in which a standard and widely-used architecture's meta-inductive bias leads to encoding statistical features rather than the low-dimensional compositional rules used to generate the task distribution. When setting out to meta-learn a structured representation using a meta-learning system, it is important to consider the meta-inductive bias of that system in addition to engineering its task distribution. Graph neural networks \citep{battaglia2018relational}, neurosymbolic approaches \citep{ellis2020dreamcoder}, as well as attention mechanisms that allow the decomposition of the input space \cite{mnih2014recurrent}, are potential meta-learning architectures with meta-inductive biases that favor structured forms of reasoning such as compositionality. ” p. 9 (Discussion)
> > > >
> > > > We would also like to clarify that a key empirical prediction from the comparison between the human and convolutional agent is not simply which does “better”, but that the pattern of behavior between the two remains quite different -- humans do better at the  compositional task than at the null task, while agents display the opposite pattern of behavior. This is unchanged when going from the feed-forward network to the convolutional one.

---

> > > > > ### Author Response · Authors · 2020-11-22
> > > > > **We have significantly reframed the paper in accordance to these comments [5/5]**
> > > > >
> > > > > > One piece of proof that the humans understand the compositional structure of the task is that they improve over the course of the task. But could this not simply reflect the fact that humans have a better capacity to improve behaviour over short time periods? In other words, how do we know that the problem with the models doesn't have to do with, say, working memory, rather than compositional understanding per se?
> > > > >
> > > > > We clarify that there are two distinct analyses that we perform for improvements in human performance. First, we consider improvement over the course of the full task in Fig 3B for inter-episode improvement. This shows that humans learn across both compositional and null tasks, and that they improve more for the compositional task. These are simply learning curves (analogous to those now also presented for the agent in the appendix). We believe this difference is driven by the differences between the two task distributions. Because humans have a bias towards learning simple, low-dimensional rules, the task distribution that was generated from such is easier for them to learn, resulting in greater improvements over the course of meta-training. Second, Fig 5B shows differences in performance over the course of a single episode. The key point we make with this analysis is that human behavior in an episode is consistent with first inferring what rules underlie the given board (low red tile hit rate) followed by a then following the deterministic production rules for that board (high red tile hit rate). This results was not intended to indicate the humans learn “better” or “faster”, but to characterize the process they are following when uncovering tiles. Agents however seem to have a closer to constant hit-rate indicating that they might not be inferring rule structure in the way humans are. For example, an agent that had a higher contant hit-rate throughout would be “better”, but still not human-like. We have clarified this with added discussion.
> > > > >
> > > > > > Altogether, this is a well put together paper. There is a lot of interesting work here, and the authors have done well to explore various facets of the setup. My main criticisms have to do with the way the work is pitched, and the way the results are interpreted. It is very much presented with a veneer that speaks to a very particular crowd in cognitive-science inspired machine learning community. But since the views in this community are not necessarily broadly shared, many of the statements, interpretations, and claims come across as quite strong and not fully corroborated.
> > > > >
> > > > > We thank the reviewer for their positive feedback and helpful constructive criticism on our framing. Their comments have led us to extensively clarify and strengthen our manuscript.

---

### Decision · Program_Chairs · 2021-01-07
**Final Decision**

**Decision:**

Accept (Poster)

**Comment:**

The authors present a study where they investigate whether meta-learning techniques leverage the underlying task distribution. To do so, the authors come up with two conditions, in the first they generate tasks using a grammar and in the second condition, which is the null condition essentially, the tasks have the same statistical properties as the compositional task but they are not derived from a simple grammar. The authors find that while humans are better in the compositional condition, models are better in the null condition.

All reviewers have been positive with this work, but some concerns were raised regarding clarity around the use of some terms, such as compositionality. The rebuttal period has been very productive and the reviewers have acknowledged the improvements on the manuscript.

All in all, I think this is a good study to appear on ICLR and I believe researchers would benefit from the design of the study that will perhaps open new opportunities around careful evaluation of meta-learning agents.